# Analysis of Fluid Balance and Urine Values in Elite Soccer Players: Impact of Different Environments, Playing Positions, Sexes, and Competitive Levels

**DOI:** 10.3390/nu16060903

**Published:** 2024-03-21

**Authors:** Jaime Sebastiá-Rico, Jose M. Soriano, Jesús Sanchis-Chordà, Ángel F. García-Fernández, Pedro López-Mateu, Sandra de la Cruz Marcos, José Miguel Martínez-Sanz

**Affiliations:** 1Area of Nutrition, University Clinic of Nutrition, Physical Activity and Physiotherapy (CUNAFF), Lluís Alcanyís Foundation-University of Valencia, 46020 Valencia, Spain; jaime.sebastia@fundacions.uv.es; 2Food & Health Lab, Institute of Materials Science, University of Valencia, 46980 Paterna, Spain; 3Joint Research Unit of Endocrinology, Nutrition and Clinical Dietetics, University of Valencia—Health Research Institute La Fe, 46026 Valencia, Spain; 4Area of Nutrition, Academia Valencia CF SAD, 46980 Paterna, Spain; 5Area of High Conditional Performance, Academia Valencia CF SAD, 46980 Paterna, Spain; 6Medical Services, Valencia CF SAD, 46980 Paterna, Spain; 7Area of Nutrition and Bromatology, University of Valladolid, 47005 Valladolid, Spain; sandra.cruz@uva.es; 8Nursing Department, Faculty of Health Sciences, University of Alicante, 03690 Alicante, Spain

**Keywords:** hydration, football, sweat rate, soccer, dehydration, fluid balance

## Abstract

Exercise can disrupt the fluid balance, hindering performance and athlete health. Limited data exist on fluid balance responses in varying climates, sexes, and ages. This study aimed to measure and compare fluid balance and urine values among elite soccer players during training at high and low temperatures, examining the differences between sexes, playing positions, and competitive levels within men’s soccer. During the 2022–2023 competitive season, a descriptive observational study was conducted on 87 soccer players from an elite Spanish soccer team. The study found that none of the groups exceeded weight loss values of 1.5% of their body mass. Additionally, the soccer players studied experienced higher weight loss, fluid intake, and a higher sweat rate (SR) during summer training compared to winter training. During the summer, male U23-21 soccer players exhibited higher levels of weight loss, fluid intake, and a higher SR compared to female soccer players or the U19-17 male category. No significant differences were found between playing positions. In conclusion, differences in the fluid balance were observed based on the climatic conditions, competitive level, and sex.

## 1. Introduction

In soccer, players cover an average total distance of approximately 8–14 km during a complete match, including 90 min of play and added extra time [1,2,3,4], characterized by a highly variable pattern of actions, such as walking, jogging, running at high and low speeds, sprinting, moving backwards, kicking, jumping, and tackling. In addition, it has been calculated that the average oxygen consumption during a match is around 70% of the maximum oxygen consumption (VO_2_ max), while the heart rate is approximately 85%, with it being rare to find values below 65% of the maximum values [4,5]. However, when soccer players are dehydrated, they experience a significantly higher heart rate, rate of perceived exertion, blood lactate level, and core body temperature [6].

Exercise can disrupt the fluid balance, which can hinder performance and athlete safety [7]. Both hyperhydration and dehydration are physiological states that should be avoided in athletes [8]. Several factors can increase the risk of dehydration or overhydration, including environmental conditions, such as humidity, altitude, and airflow; fluid availability, such as water stations and hydration breaks; exercise structure, such as the number of daily sessions, duration, and intensity of the exercise; and athlete factors, such as gender, acclimatization status, and body composition; as well as factors specific to the sport being played [7,9,10].

Advance planning through an understanding of each player’s individual sweat loss patterns, customized hydration plans, and education on hydration can help ensure that each player receives the appropriate amount of fluid to meet their needs [4]. It is important to note that the sensation of thirst may not always accurately reflect the body’s fluid needs [11]. Additionally, the high-intensity nature of soccer, coupled with limited fluid availability during competitions, may increase the risk of dehydration for players [4,12].

It is commonly accepted that during exercise lasting for about two hours or less, the amount of water lost through respiration in the lungs is trivial, and the associated changes in metabolic mass can be ignored [13,14]. Therefore, it is often assumed that for a soccer player, every 1 kg of body mass lost after physical exertion represents approximately 1 L of water loss. This makes it easy to calculate the amount of sweat lost by measuring the change in body mass after correcting for fluid intake [15,16]. Studies indicate that a 2% loss of body weight can negatively impact athletic performance [8], as well as increase the likelihood of heat-related illness [11] and injury [17]. Other methods of assessing hydration levels besides body weight change include thirst level, urine color, urine osmolality, sweat rate (SR), sweat analysis, and urine specific gravity (USG) [4,18,19]. Consequently, the athlete should try to reach the state of euhydration, which represents the optimal water balance in the human body, essential for metabolic function, thermoregulation, and general health, in order to achieve optimal athletic performance and minimize thermoregulatory and physiological stress [11,20].

Female athletes typically have a lower SR than males because they have less muscle tissue to contribute to metabolic heat production during exercise, as well as lower absolute work rates, which could affect their water requirements compared to males [21,22]. In addition, different playing positions have varying anthropometric characteristics and physical demands during matches. Therefore, hydration needs may differ between matches and training [23,24,25,26]. Finally, it is important to consider that total daily fluid requirements may vary with age. Adolescents, due to their larger body surface area relative to their mass, may experience greater heat stress when exercising in hot environments compared to adults. This is because they absorb ambient heat more readily than adults [9,11].

In summary, a soccer player’s water requirements are influenced by various factors, such as gender, age, daily physical activity, and the environment in which they are playing. However, published studies evaluating parameters related to fluid status in soccer players have not compared all of these characteristics in the same study. Therefore, the aim of this study was to measure and compare the SR, fluid balance, and urine values in elite soccer players during training at high and low temperatures, looking for differences between sexes, playing positions, and competitive levels within men’s soccer. Accordingly, it was initially hypothesized that:

**Hypothesis (H1).** 
*During training, fluid intake will be insufficient according to the official recommendations and calculated SR.*


**Hypothesis (H2).** 
*The SR and fluid intake will be higher when training at higher temperatures.*


**Hypothesis (H3).** 
*For physiological reasons, there will be significant differences in the fluid balance, SR, and urine values between the sexes (female team and U23-21) and male competition levels (U23-21 and U19-17). However, there will be no significant differences between playing positions.*


## 2. Materials and Methods

### 2.1. Study Design

This is a descriptive observational study on the fluid balance, SR, and urine values in elite soccer players of both sexes, who belong to the Valencia C.F. Academy. The sample size calculation was performed with the RStudio software (version 3.15.0, RStudio Inc., Boston, MA, USA). The significance level was set a priori at *p* = 0.05. The standard deviation (SD) was set according to the total sweat rate (L/h) data from previous studies on male elite Spanish soccer players (DE = 0.21) [27]. With an estimated error (d) of 0.057, the sample size needed was 62 soccer players. The study population was selected by non-probabilistic, non-injury, convenience sampling among elite soccer players of both sexes belonging to the Valencia C.F. Academy. In addition, the study design, as well as the development of the manuscript, followed the STROBE statement [28].

### 2.2. Participants

The study included all soccer players from the Valencia Mestalla (U23-21), Juvenil A (U19-17), and Valencia CF Femenino teams. However, some players did not return to the club after the first trial, while others were unavailable due to sports-related injuries or promotion to the first team. Therefore, 19 players from the U23-21, 21 from the U19-17, and 22 from Valencia CF Femenino completed the warm environment training, while 12 from the U23-21, 6 from the U19-17, and 7 from Valencia CF Femenino completed the cold environment training. Thus, the sample was analyzed by dividing it into two subgroups: (1) the sample with repeated measurements (RM), and (2) the sample with non-repeated measurements (n-RM). All athletes have competed for at least nine years and perform five regular training sessions, along with two additional sessions per week, lasting approximately 90 to 120 min per day. They play one official soccer match per week. Women who were menstruating were excluded from the study, as this situation may affect urine samples. Table 1 shows the characteristics of the players, differentiating between players who took the test in both weather conditions and players who only participated in one measurement.

### 2.3. Procedure

The assessment was conducted during two training sessions in the month of January and August, during the competition season 2022–2023 in Paterna, Valencia (Spain). One session occurred in the winter (cool), while the other took place in the summer (warm) (Table 2). According to the coach’s guidelines, the intensity of the training was considered to be high (MD-3) and was similar in both cases. The sessions were representative of those normally performed by the team on typical days of the week and times in the season. The training sessions consisted of a 20 min warm-up on both the field and in the gym, followed by intervals of running, ball exercises, and 100 min soccer matches. All training sessions took place between 10:00 a.m. and 12:30 p.m., and all players followed the same training protocol. The players wore standard training and match equipment, including shorts, T-shirts, sports bras (for women), socks, shin guards, and soccer boots. The coach conducted the training without any interference from the research team. The players were not required to undergo any dietary control as part of the research. However, they were asked to urinate and defecate prior to the pre-training measurements. The club’s nutritionist insisted that the players follow the general hydration and nutrition protocol the day before the sample collection to ensure that all the players were well hydrated before the test.

The athletes were instructed to drink only from their assigned flasks. They were informed to notify the researcher when their flask was empty and needed to be refilled. After the training, the liquid from the canisters was poured into a beaker. The amount of liquid consumed by the athlete during training was calculated by subtracting the volume of liquid poured into the beaker from the total volume of liquid in the canister. To measure the volume of urine excreted by the soccer players, we recorded the volume of urine from the start of each training session until the end. For this purpose, soccer players may urinate before the pre-training weigh-in without the need to measure the volume of urine excreted. However, after the initial weigh-in, they must use an aseptic container with a 2 L capacity for urine collection. The protocol complies with the Declaration of Helsinki for human research and is approved by the Ethics Committee at the University of Valencia (1534145).

### 2.4. Instruments

The kinanthropometric measurements were conducted in accordance with ISO 7250-1:2017 [29] and the International Society for the Advancement of Kynanthropometry (ISAK) standard [30], by an accredited ISAK 1 anthropometrist. The body mass and height measurements were taken using a Seca 515/514 bioimpedance machine with an accuracy of 50 g and a Seca 213 transportable stadiometer with an accuracy of 1 mm, respectively. The soccer players were weighed in their underwear and barefoot before the start of training. They were instructed to urinate or defecate before the initial weigh-in. After training, the athletes wiped sweat from their faces, torso, and legs before being weighed again. To record the temperature, relative humidity, and wind speed at the research site, we utilized data from the Agencia Estatal de Meteorología (AEMET) to record the meteorological parameters. These environmental parameters were recorded from the warm-up’s start until the end of the training sessions and documented on a record sheet.

The percentage of weight lost was calculated using the equation proposed by Martins [31]: Percentage (%) Weight Lost or Dehydration = [(Weight before − Weight after)/Weight before] − 100. Weight expressed in kg. The calculation of the SR was carried out using the following formula, taken from Murray [19]: SR = (Weight Lost + Fluid Ingested − Urine)/Minutes Activity. Weight expressed in kg.

After the training session, the hydration status was estimated by the urine specific gravity (USG), measured with a digital refractometer (REC-200ATC, Gain Express Holdings Ltd., Hong Kong, China), and the urine color, using the Armstrong scale of 1 to 3 hydrated, 4 to 6 dehydrated, and 7 or 8 seriously dehydrated [32]. The hydration status using the USG was classified as follows: hydrated (USG ≤ 1.020), dehydrated (USG 1.021–1.024), and seriously dehydrated (USG > 1.024) [20]. The urine color was analyzed in all players who were able to urinate before training; all of them being in a hydrated state.

### 2.5. Stadistical Analysis

The normal distribution of the variables was evaluated using the Kolmogorov–Smirnov and Shapiro–Wilk tests. The homogeneity of variances was analyzed with the Levene test. Categorical variables are described as absolute (n) and relative (%) frequencies, and quantitative variables as the mean (SD). Comparisons between the parametric quantitative variables of 2 independent samples (sample of subjects with one measurement) were made with Student’s *t*-test, while the Mann–Whitney U test was employed for the non-parametric ones. Depending on the normality of the variables, comparisons of the quantitative variables of paired samples (sample of subjects with two measurements) were evaluated using Student’s *t*-test for repeated measurements or the Wilcoxon W test. The comparison between more than 2 independent means of parametric variables was analyzed by the variance analysis (ANOVA) test; however, the Kruskal–Wallis test was employed when the variables did not follow normal distribution. The comparison between more than 2 means of parametric variables from the sample with repeated measurements were studied using the ANOVA test for repeated measures or the Friedman test if the variables were non-parametric. Differences in the categorical variables were evaluated with the chi-square test. Statistical significance was set at *p* < 0.05. Statistical analysis was performed with the statistical package SPSS 26.0 for Windows.

## 3. Results

### 3.1. Fluid Balance and Sweat Rate

Table 3 shows the fluid balance, SR, and USG values for the two samples analyzed by sex, and in both winter and summer training. During summer training, U23-21 players experienced higher weight loss (RM: 0.87 ± 0.59 kg; 1.17 ± 0.82%/n-RM: 0.96 ± 0.63 kg; 1.27 ± 0.8%), fluid intake (RM: 1.97 ± 0.88 L/n-RM: 2.12 ± 0.73 L), SR (RM: 1.34 ± 0.46 L/h/n-RM:1.46 ± 0.40 L/h), and USG in RM (1.03 ± 0.01) compared to winter training. However, there were no statistically significant differences in the urine output. Female players presented a significantly higher SR during summer training (RM: 0.72 ± 0.17 L/h/n-RM: 0.68 ± 0.16 L/h) in comparison with winter training (RM: 0.63 ± 0.14 L/h/n-RM: 0.55 ± 0.10 L/h). When compared by sex, males had higher weight loss (RM: 0.87 ± 0.59 kg; 1.17 ± 0.82%/n-RM: 0.96 ± 0.63 kg; 1.27 ± 0.8%), fluid intake (RM: 1.97 ± 0.88 L/n-RM: 2.12 ± 0.73 L), SR (RM: 1.34 ± 0.46 L/h/n-RM:1.46 ± 0.40 L/h), and USG (RM: 1.03 ± 0.01/n-RM: 1.02 ± 0.01) during summer training, as well as in the total sample. No statistically significant differences were observed between playing positions during summer and winter training in n-RM. However, midfielders showed higher weight loss (1.07 ± 0.68 kg/1.56 ± 0.90%), fluid intake (1.96 ± 0.82 L), and SR (1.39 ± 0.55 L/h) in the warm training session and lower USG (1.02 ± 0.01) than the other playing positions in the RM (Appendix A).

Table 4 shows the hydration values and fluid balance between the competitive levels for men in both winter and summer training. U19-17 athletes had a higher SR (RM: 1.18 ± 0.31 L/h/n-RM: 1.09 ± 0.25 L/h), fluid intake (RM: 1.72 ± 0.28 L/n-RM: 1.76 ± 0.37 l), and weight loss in RM (0.82 ± 0.62 kg/1.15 ± 0.86%) during warm training, while the USG in n-RM was higher during cool training (1.03 ± 0.00). No statistically significant differences were found in the urine output. When comparing the two levels of competition, U23-21 players showed a higher weight loss (U23-21: 0.96 ± 0.63 kg/U19-17: 0.58 ± 0.53 kg), fluid intake (U23-21: 2.12 ± 0.73 L/U19-17: 1.76 ± 0.37 L), and SR (U23-21: 1.46 ± 0.40 L/h/U19-17: 1.09 ± 0.25 L/h) during summer training in n-RM.

Figure 1 shows the SR values by team and by season of the year in which the evaluation was carried out. During the summer training, the female team had the lowest SR value (RM: 0.72 ± 0.17 L/h/n-RM: 0.68 ± 0.16 L/h), followed by the U19-17 (RM: 1.18 ± 0. 31 L/h/n-RM: 1.09 ± 0.25 L/h), while the U23-21 (RM: 1.34 ± 0.46 L/h/n-RM: 1.46 ± 0.40 L/h) had the highest value (RM: 1.34 ± 0.46 L/h/n-RM: 1.46 ± 0.40 L/h). In addition, the female team exhibited the lowest mean SR (n-RM: 0.65 ± 0.16 L/h), followed by the U19-17 (n-RM: 0.98 ± 0.33 L/h), and the U23-21 (n-RM: 1.14 ± 0.54 L/h).

### 3.2. Hydration Status

Table 5 shows the hydration status as a function of sex in both winter and summer training. No statistically significant differences in the urine color status and USG were observed, regardless of the type of training. No statistically significant differences were observed between the playing positions during summer and winter training (Appendix A).

Table 6 shows the hydration status as a function of the level of competition in both winter and summer training. Statistically significant differences in the urine color were observed during winter training in n-RM, with the entire sample of the U23-21 team in the DH category. No statistically significant differences in the USG and urine color were observed during summer training. Figure 2 shows the hydration status by team and by season of the year in which the evaluation was carried out.

## 4. Discussion

To our knowledge, this is the first study to analyze the fluid balance, SR, and urine values during training in different climatic conditions in elite soccer players, comparing them across playing positions, sexes, and male competitive levels. The main results were as follows: (1) neither group consumed the recommended amount of fluids based on the official recommendations and calculated SR, but they did not exceed weight loss values of 1.5% of their body mass; (2) weight loss, fluid intake, and SR were higher during summer training than winter training in all the groups of soccer players studied. Nonetheless, urine collected did not vary. (3) During summer, U23-21 players showed higher values for weight loss, fluid intake, and SR than female athletes, or the U19-17 male category; (4) no differences in the hydration status or SR were observed between playing positions.

### 4.1. Fluid Balance and Sweat Rate

The International Olympic Committee (IOC) recommends, as a general guideline, the consumption of 6 mL of fluid per kg of body mass every 2–3 h, which amounts to 2–3 L per day. Additionally, it is recommended to consume 1–2 L of fluid per hour of exercise and 1 L for every 5 °C increase in ambient temperature above 21.5 °C [33]. However, the SR of athletes varies significantly due to factors such as environmental conditions, metabolic rate, clothing, and heat acclimatization [7,33]. The amount of fluid lost through sweating can range from less than 1 to more than 3 L per hour [33]. Therefore, it is difficult to establish an absolute recommendation for fluid intake during exercise. Additionally, limiting the rate of gastric emptying affects the amount of fluid an athlete can ingest during competition [33].

In a recent study, data were collected from different sports modalities, where 268 soccer players were included, obtaining mean values of 0.94 ± 0.38 L/h in SR and 795 ± 441 mg/h (34.6 ± 19.2 mmol/h) in lost sodium [34]. However, it should be noted that they did not differentiate between level categories or sexes. In a second study, they included 497 soccer players (415 male and 82 female) and found a mean SR range of 0.3–2.5 L/h, although, as in the previous study, the data were not differentiated between athletes [8]. In addition, the review assessed the potential impact of hypohydration on cognitive function, sport-specific skills, sprinting ability, and high-intensity intermittent performance in soccer. The data indicate significant negative effects; however, additional research is required for more precise conclusions. The average results for the SR in the U23-21 category (1.14 ± 0.54 L/h), both in summer (RM: 1.34 ± 0.46 L/h/n-RM: 1.46 ± 0.40 L/h) and winter (RM: 0.55 ± 0.16 L/h/n-RM: 0.62 ± 0.21 L/h), are consistent with the previously mentioned reviews, thus supporting the consistency of our findings.

In relation to weight lost, Shirreffs et al. evaluated fluid intake (830 ± 380 mL) and weight loss (1.5 ± 0.5%) after a 90 min training session in 67 soccer players, from three European professional teams [35]. However, the study did not find any relationship between body mass and sweat loss or volume of fluid consumed. Therefore, factors such as activity rate, heat acclimatization status, and genetic differences likely contribute to the large variability observed among soccer players. Another study collected data from 19 semi-professional soccer players on their SR (0.86 ± 0.42 L/h), fluid intake (768 ± 292 mL), body weight loss (1.07 ± 0.62 kg and 1.46 ± 0.84%), and environmental conditions (23.9 ± 2.63 °C) over an 11-week period [36]. Although most soccer players did not match fluid loss during training, the results showed that a fluid intake above ~10 mL·kg^−1^ body weight was associated with an increase in the total distance traveled and high-speed running distance during prolonged training sessions (>110 min) [36].

The weight loss values observed in our study were lower than those reported in the literature. The mean weight loss values for U23-21 soccer players were 1.05 ± 0.76%, for U19-17 players it was 0.74 ± 0.81%, and for female players it was 0.60 ± 0.67%. Our results found that fluid intake during training was higher than previously reported. Specifically, the mean values for U23-21 (1.65 ± 0.86 L), U19-17 (1.61 ± 0.53 L), and female players (1.13 ± 0.49 L) were all higher than in previous studies. This trend was also observed in both summer and winter values across all three groups of soccer players. Possible reasons for the observed differences between players may include environmental factors such as temperature or humidity, variations in training intensity, differences in acclimatization status, body composition (including fat and muscle mass), and variations in dietary education and hydration habits.

When quantifying a player’s hydration, it is important to consider training intensity. Several studies have shown that higher intensity training is associated with greater weight loss, greater loss of sodium in sweat, and a higher SR [27,37]. Therefore, the hydration needs of soccer players should be increased accordingly. Our research was conducted during MD-3 in both summer and winter to ensure that all soccer players were subjected to the same physical demands and to eliminate bias.

Women experience changes in fluid balance throughout the menstrual cycle due to hormonal variations. In the follicular phase, estrogen, luteinizing hormone, and follicle-stimulating hormone levels increase, affecting the regulation of body fluids. During the luteal phase, there is a significant increase in body temperature of 0.3 to 0.5 °C at rest and during exercise. This could increase the risk of heat-related illness, especially if combined with dehydration [38,39]. Thus, in female elite soccer players, it was observed that sweat sodium loss and SR during exercise were related to basal testosterone concentration, but not to basal cortisol concentration or the testosterone/cortisol ratio [40]. Hormonal changes can impact fluid dynamics, body temperature, thirst, and, consequently, athletic performance and overall health [8,38]. Nevertheless, it is important to note that the specific interaction between SR, the menstrual cycle, and hydration requires further research to fully understand its impact on women’s health and performance.

Studies have shown that women generally have lower SR values due to their lower body mass and lower absolute work rates [12,21]. Our results are consistent with the scientific literature, indicating lower values in female soccer players (0.65 ± 0.16 L/h) compared to U23-21 male soccer players (1.14 ± 0.54 L/h), particularly during summer training in both RM (0.72 ± 0.17 L/h vs. 1.34 ± 0.46 L/h) and n-RM (0.68 ± 0.16 L/h vs. 1.46 ± 0.40 L/h).

A recent study assessed the fluid balance of eight female professional soccer players during a training session and a match [12]. During training, the athletes experienced a slight increase in weight (0.2 ± 0.3 kg; 0.29 ± 0.57%) and a urine output of 104 ± 257 mL, along with a SR of 0.49 ± 0.26 L/h. In contrast, during the match, the athletes experienced a decrease in weight (0.7 ± 0.5 kg; 1.12 ± 0.86%) and a urine output of 107 ± 83 mL, along with a SR of 0.85 ± 0.30 L/h [12]. Our study found that weight decreased after training (0.35 ± 0.41 kg; 0.60 ± 0.67%), while the urine output (0.20 ± 0.15 L) and SR (0.65 ± 0.16 L/h) remained similar.

Concerning young soccer players, there seems to be a tendency to underestimate their sweat losses, which can lead to inadequate hydration during training (especially in hot conditions) [41]. Food education is crucial for optimizing the nutrition and hydration of young soccer players. It can enhance their knowledge and enable them to apply it during their future careers. Effective interventions include talks, informative posters, infographics, and web or mobile applications. These interventions have been shown to improve knowledge and habits in young soccer players [42,43,44]. In our study, we found that U19-17 soccer players had lower values for SR, fluid intake, and weight loss after exercise compared to U23-21 soccer players, particularly during summer training. Physiological differences may contribute to discrepancies in these values. Factors such as a higher basal metabolic rate in adults, a higher body mass to body surface area ratio, more efficient thermoregulatory capacity in adults, physiological development, and established hydration habits may result in a greater need for thermoregulation through sweating during exercise [4,42,45].

### 4.2. Hydration Status and Environment

The prevalence of starting exercise in a hypohydrated state has been observed in soccer players. The prevalence of hypohydration was observed to be 63.3% using the USG as a measurement method, 37.4% using urine osmolarity, and 58.8% using urine color [45]. Male, elite soccer players during pre-training had a higher prevalence of hypohydration states (66.0%, 66.2%, and 79.6%, respectively) compared to female, recreational, and pre-game players (49.4%, 55.6%, and 41.3%, respectively) [45]. Furthermore, the scientific literature documents instances of young athletes arriving at training sessions and competitions already dehydrated [4].

Exercise in hot conditions led to significant increases in heart rate, the rate of perceived exertion, and skin temperature, while the temperature gradient decreased [33,46]. Conversely, no significant changes in blood lactate concentration were observed during exercise in cold conditions [46]. These results suggest that the environmental temperature has a significant impact on the physiological response during exercise, which may affect perceived fatigue and performance. During prolonged exercise in hot environmental conditions, the IOC recommends consuming fluids containing diluted carbohydrates and electrolytes to improve the overall fluid intake. On the one hand, the diluted carbohydrates and electrolytes increase the palatability of fluids [33]. On the other hand, carbohydrate intake during exercise can increase the time to exhaustion and decrease the heart rate during tests [46]. None of the soccer players studied consumed carbohydrate-rich beverages during training. This may be due to personal preference or nutritional recommendations from the nutritionist based on the caloric and/or carbohydrate load of the microcycle, which focuses on carbohydrate periodization [47].

Regarding low temperatures, they may lead to reduced hydration needs. However, the relationship between sweat loss and fluid intake is complex and can vary between individuals. Additionally, in these conditions, athletes may experience a decreased thirst sensation, which could increase the risk of dehydration if proper hydration habits are not maintained [14]. A study was conducted to analyze the fluid and electrolyte balance of 17 professional soccer players during a training session at 5.1 ± 0.7 °C. The players experienced a weight loss of 1.62 ± 0.55% (1.27 ± 0.47 kg), their water intake was 423 ± 215 mL, and their SR was 1.13 ± 0.30 L/h [14]. Another study assessed the fluid balance of 14 professional soccer players in response to low and high cold training intensities. The study found that the SR (0.98 ± 0.21 L/h) and fluid intake (505 ± 265 mL) were higher at a high intensity compared to a low intensity (0.55 ± 0.20 L/h and 394 ± 160 mL, respectively), while maintaining a similar USG (1.023 ± 0.004 at high intensity and 1.024 ± 0.005 at low intensity) [27].

Considering the IOC’s recommendation to keep USG levels below 1.020 [33], our results showed that most of the players in all categories ended up dehydrated or severely dehydrated, especially the U19-17 in winter training (RM: 1.03 ± 0.01/n-RM: 1.03 ± 0.00). There could be several reasons for this. Firstly, it may be due to inadequate hydration or poor dietary intake prior to training, although it should be noted that the club’s nutritionist provided and implemented the necessary dietary and hydration recommendations to the players the day before the test. Additionally, it could be attributed to the intensity of the training, as it was demanding for all players. However, it is important to note that these players are considered elite and have at least 9 years of competitive experience.

In addition to testing the USG, another method to assess the hydration of soccer players is through bioimpedance (BIA). BIA is a non-invasive and easy-to-apply method based on the principle that body water conductivity varies between different compartments and can be used to calculate fat mass and fat-free mass (including total body water) [48]. Determining the water status prior to training and/or competitions could enhance hydration patterns and prevent adverse effects, such as dehydration and hyperhydration, which can negatively impact sports performance or increase the risk of injury [49]. However, it is important to note that certain diseases, treatments, clinical situations, and individual conditions (such as ethnicity and extremity position), as well as the type of bioimpedance (including the number of electrodes and frequency), and rules of use, may affect the results of this instrument [50]. Therefore, it is advisable to combine methods to avoid measurement errors.

### 4.3. Limitations

This research had some limitations that should be discussed. Although the study population was small, all Spanish soccer players in the selected sample were recruited, which is a significant sample size according to the statistical principles applied. In addition, the concentrations of electrolytes, such as sodium or potassium, in sweat were not analyzed due to logistical reasons, which is useful for the quality control of sweat samples. Another limitation of the present study was that fluid intake was only assessed during exercise. Assessing the first morning USG, 24 h urine volumes, and/or 24 h fluid intake before training would provide information on the influence of basal hydration on the SR and urine color. Additionally, the second evaluation was completed 5 months after the initial tests conducted in August. It is unclear whether the sweat rate and fluid intake of the players were influenced by acclimatization, food education by the club dietitian-nutritionist, or changes in their physical condition during the observed period. In addition, none of the players consumed carbohydrates during exercise, and future studies could explore how carbohydrate sources, such as gels, bars, and sports drinks, may affect players’ ad libitum water intake. Lastly, although the sample incorporating n-RM is larger, it is important to consider both intra- and inter-individual variability when interpreting the results. However, we included the RM sample to enhance the sensitivity and reliability of the results.

### 4.4. Practical Applications

The following practices are recommended for soccer clubs due to the variability of fluid balance and urine values:Food education: Implement food education programs for soccer players, especially in youth categories, to promote understanding about and the importance of adequate fluid and food intake to support sports performance and recovery. The use of infographics, presentations, or posters can contribute to this. In addition, it is advisable to encourage the consumption of seasonal foods that can contribute to hydration, such as vegetables like beet, cucumber, and tomato, and fruits like watermelon, orange, and melon, as well as animal milk or vegetable drinks.Body composition monitoring: To monitor changes in hydration and adjust nutritional and hydration strategies, body composition assessment methods, such as electrical bioimpedance, should be used.Sex: It is important to consider sex differences when it comes to hydration. Recognize and address differences in hydration needs between males and females, including the influence of the menstrual cycle on the sweat rate and sodium loss.Climate adaptation: To prepare athletes for competing in high temperature conditions and minimize the risk of heat-related illnesses, climate adaptation programs should include acclimatization to different temperatures and relative humidity.Urinary values: Urinary indices can be used as a tool to assess the hydration status of athletes before and after physical demands. This information can be used to make timely adjustments in fluid intake.Hydration interventions during training and competitions: It is recommended to establish hydration practices and protocols during exercise sessions. This includes providing regular drinking breaks and making hypotonic, isotonic, or hypertonic beverages available, according to individual needs.Individualized hydration strategies: It is crucial to develop individualized hydration plans for athletes, since hydration affects cognitive, technical, and physical performance. To achieve this, it is desirable to know certain values, such as the sweat rate or sodium concentration in sweat, not only at the team level but also on an individual basis.

## 5. Conclusions

The main findings indicate that the soccer players did not meet the recommended fluid intake based on the official guidelines and calculated SR; however, they did not exceed a 1.5% body mass loss. Furthermore, summer training sessions consistently resulted in higher weight loss, fluid intake, and SR compared to winter sessions across all player categories. Among the U23-21 male football players, these values were higher than those of their female counterparts and the U19-17 male category. Lastly, no significant differences were observed between the various playing positions.

## Figures and Tables

**Figure 1 nutrients-16-00903-f001:**
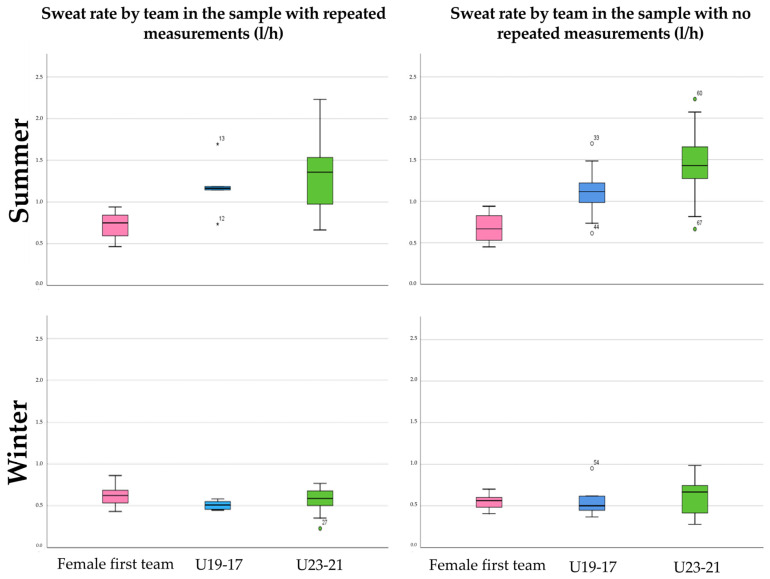
Sweat rate by team.

**Figure 2 nutrients-16-00903-f002:**
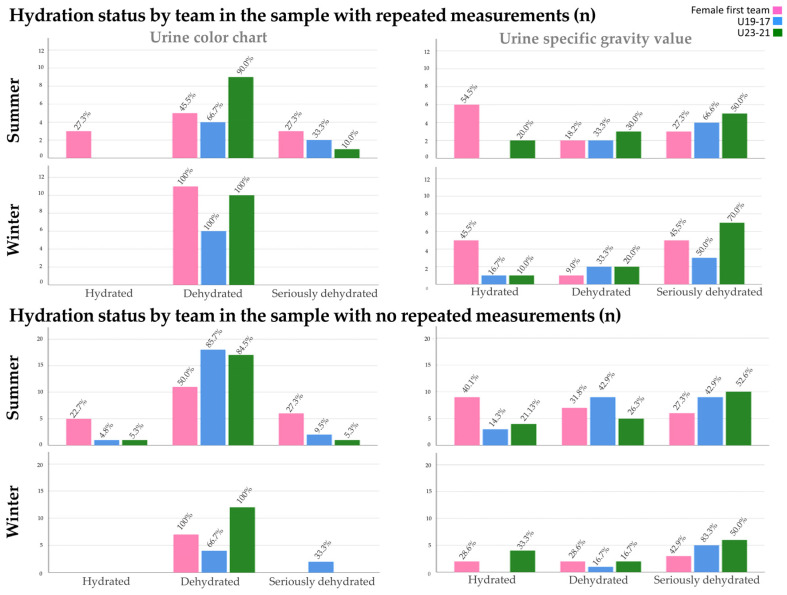
Hydration status by team.

**Table 1 nutrients-16-00903-t001:** Characteristics of the sample.

**RM Sample**									
	Males (n = 16)			Females (n = 11)			Total (n = 27)		
Age (years)	19.9 (2.4)			22.6 (3.5)			21.0 (3.15) *		
Weight (kg)	72.9 (4.1)			61.6 (8.7)			68.3 (8.4) *		
Height (cm)	181.2 (6.0)			165.0 (5.9)			173.1 (8.1) *		
**n-RM sample**									
	Males			Females			Total		
	W (n = 40)	C (n = 18)	T (n = 58)	W (n = 22)	C (n = 7)	T (n = 29)	W (n = 62)	C (n = 25)	T (n = 87)
Age (years)	18.4 (2.4)	19.2 (2.9)	18.7 (2.5)	21.0 (3.7)	25.0 (4.8)	22.0 (4.3) ^†^	19.3 (3.1) *	20.8 (4.3) *	19.8 (3.6) *
Weight (kg)	70.8 (6.7)	72.6 (7.3)	71.3 (6.8)	59.0 (7.8)	63.5 (4.8)	60.1 (7.4)	66.6 (9.0) *	70.0 (7.8) *	67.6 (8.8) *
Height (cm)	181.3 (6.2)	181.7 (6.9)	181.5 (6.5)	165.4 (5.7)	165.4 (5.9)	165.4 (5.8)	173.4 (7.9) *	173.6 (8.1) *	173.5 (8.0) *

W: summer measurement; C: winter measurement; T: total. Variables are expressed as mean (SD). * *p* < 0.05 males vs. females. ^†^ *p* < 0.05 warm vs. cool.

**Table 2 nutrients-16-00903-t002:** Weather conditions during the three training sessions of testing.

	U23-21		U19-17		Female	
	C	W	C	W	C	W
Initial temperature (°C)	6	26	12	23	13	23
Final temperature (°C)	10	29	15	29	18	27
Initial relative humidity (%)	36	77	88	44	82	49
Final relative humidity (%)	28	59	72	35	51	37
Initial wind speed (km·h^−1^)	12	6	14	11	11	16
Final wind speed (km·h^−1^)	6	12	12	22	4	11

W: summer measurement; C: winter measurement.

**Table 3 nutrients-16-00903-t003:** Hydration and fluid balance values of the sample, as a function of gender.

	Female			U23-21		
	W	C	T	W	C	T
**RM sample**						
n			11			10
Weight loss:						
kg	0.40 (0.52)	0.31 (0.39)	-	0.87 (0.59) *	0.42 (0.45)	-
%	0.61 (0.80)	0.48 (0.59)	-	1.17 (0.82)	0.53 (0.58)	-
Fluid intake (L)	1.32 (0.57)	1.10 (0.40)	-	1.97 (0.88) *^†^	0.83 (0.44)	-
Urine output (L)	0.27 (0.17)	0.15 (0.11)	-	0.15 (0.08)	0.14 (0.08)	-
SR:						
L/h	0.72 (0.17) ^†^	0.63 (0.14)	-	1.34 (0.46) *^†^	0.55 (0.16)	-
mL/min	12.05 (2.81) ^†^	10.46 (2.37)	-	22.38 (7.67) *^†^	9.24 (2.75)	-
USG	1.02 (0.01) ^†^	1.02 (0.01)	-	1.03 (0.01) *^†^	1.03 (0.00) *	-
**n-RM sample**						
n	22	7	29	19	12	31
Weight loss:						
kg	0.38 (0.43)	0.29 (0.39)	0.35 (0.41)	0.96 (0.63)	0.52 (0.40)	0.79 (0.59) ^†^
%	0.63 (0.69)	0.48 (0.66)	0.60 (0.67)	1.27 (0.8)	0.69 (0.49)	1.05 (0.76) ^†^
Fluid intake (L)	1.20 (0.47)	0.92 (0.54)	1.13 (0.49)	2.12 (0.73)	0.91 (0.39)	1.65 (0.86) ^†^
Urine output (L)	0.23 (0.16)	0.11 (0.05)	0.20 (0.15)	0.16 (0.08)	0.19 (0.12)	0.17 (0.09)
SR:						
L/h	0.68 (0.16)	0.55 (0.10)	0.65 (0.16) ^†^	1.46 (0.40)	0.62 (0.21)	1.14 (0.54) ^†^
mL/min	11.28 (2.65)	9.12 (1.69)	10.77 (2.60) ^†^	24.39 (6.73)	10.27 (3.49)	18.9 2 (8.97) ^†^
USG	1.02 (0.01)	1.02 (0.00)	1.02 (0.01)	1.02 (0.01)	1.02 (0.01)	1.02 (0.01)

W: summer measurement; C: winter measurement; SR: sweat rate; T: total; USG: urine specific gravity; variables are expressed as mean (SD). * *p* < 0.05 males vs. females. ^†^ *p* < 0.05 warm vs. cool.

**Table 4 nutrients-16-00903-t004:** Hydration values and fluid balance of the sample, as a function of male competitive level.

	U19-17			U23-21		
	W	C	T	W	C	T
**RM sample**						
n			6			10
Weight loss:						
kg	0.82 (0.62) ^†^	0.27 (0.21)	-	0.87 (0.59)	0.42 (0.45)	-
%	1.15 (0.86) ^†^	0.37 (0.27)	-	1.17 (0.82)	0.53 (0.58)	-
Fluid intake (L)	1.72 (0.28) ^†^	0.88 (0.14)	-	1.97 (0.88) ^†^	0.83 (0.44)	-
Urine output (L)	0.18 (0.08)	0.14 (0.04)	-	0.15 (0.08)	0.14 (0.08)	-
SR:						
L/h	1.18 (0.31) ^†^	0.51 (0.05)	-	1.34 (0.46) ^†^	0.55 (0.16)	-
mL/min	19.69 (19.41) ^†^	8.45 (0.88)	-	22.38 (7.67) ^†^	9.24 (2.75)	-
USG	1.03 (0.00)	1.03 (0.01)	-	1.03 (0.01)	1.03 (0.00)	-
**n-RM sample**						
n	21	6	27	19	12	31
Weight loss:						
kg	0.58 (0.53)	0.19 (0.66)	0.50 (0.57)	0.96 (0.63)	0.52 (0.40)	0.79 (0.59) ^†^
%	0.87 (0.78)	1.12 (0.83)	0.74 (0.81)	1.27 (0.8)	0.69 (0.49)	1.05 (0.76) ^†^
Fluid intake (l)	1.76 (0.37)	1.08 (0.68)	1.61 (0.53) ^†^	2.12 (0.73)	0.91 (0.39)	1.65 (0.86) ^†^
Urine output (l)	0.15 (0.09)	0.15 (0.09)	0.15 (0.09)	0.16 (0.08)	0.19 (0.12)	0.17 (0.09)
SR:						
L/h	1.09 (0.25)	0.56 (0.21)	0.98 (0.33) ^†^	1.46 (0.40)	0.62 (0.21)	1.14 (0.54) ^†^
mL/min	18.24 (4.19)	9.38 (3.45)	16.27 (5.47) ^†^	24.39 (6.73)	10.27 (3.49)	18.9 2 (8.97) ^†^
USG	1.02 (0.01)	1.03 (0.00)	1.03 (0.00) ^†^	1.02 (0.01)	1.02 (0.01)	1.02 (0.01)

W: summer measurement; C: winter measurement; SR: sweat rate; T: total; USG: urine specific gravity; variables are expressed as mean (SD). ^†^ *p*<0.05 warm vs. cool.

**Table 5 nutrients-16-00903-t005:** Hydration status as a function of sex.

		Female		U23-21	
		CO	USG	CO	USG
**RM sample. (n (%))**					
n		11		10	
H:	W	3 (27.3)	6 (54.5)	0 (0.0)	2 (20.0)
	C	0 (0.0)	5 (45.5)	0 (0.0)	1 (10.0)
DH:	W	5 (45.5)	2 (18.2)	9 (90.0)	3 (30.0)
	C	11 (100.0)	1 (9.1)	10 (100.0)	2 (20.0)
SDH:	W	3 (27.3)	3 (27.3)	1 (10.0)	5 (50.0)
	C	0 (0.0)	5 (45.5)	0 (0.0)	7 (70.0)
**n-RM sample. (n (%))**					
n		29		31	
H:	W	5 (22.7)	9 (40.1)	1 (5.3)	4 (21.1)
	C	0 (0.0)	2 (28.6)	0 (0.0)	4 (33.3)
DH:	W	11 (50.0)	7 (31.8)	17 (84.5)	5 (26.3)
	C	7 (100.0)	2 (28.6)	12 (100.0)	2 (16.7)
SDH:	W	6 (27.3)	6 (27.3)	1 (5.3)	10 (52.6)
	C	0 (0.0)	3 (42.9)	0 (0.0)	6 (50.0)

W: summer measurement; C: winter measurement; CO: urine color; DH: dehydrated; H: hydrated; SDH: seriously dehydrated; USG: urine specific gravity. Variables are expressed as n (%).

**Table 6 nutrients-16-00903-t006:** Hydration status as a function of the male competitive level.

		U19-17		U23-21	
		CO	USG	CO	USG
**RM sample. (n (%))**					
n		6		10	
H:	W	0 (0.0)	0 (0.0)	0 (0.0)	2 (20.0)
	C	0 (0.0)	1 (16.7)	0 (0.0)	1 (10.0)
DH:	W	4 (66.7)	2 (33.3)	9 (90.0)	3 (30.0)
	C	6 (100.0)	2 (33.3)	10 (100.0)	2 (20.0)
SDH:	W	2 (33.3)	4 (66.7)	1 (10.0)	5 (50.0)
	C	0 (0.0)	3 (50.0)	0 (0.0)	7 (70.0)
**n-RM sample. (n (%))**					
n		27		31	
H:	W	1 (4.8)	3 (14.3)	1 (5.3)	4 (21.1)
	C	0 (0.0)	0 (0.0)	0 (0.0) *	4 (33.3)
DH:	W	18 (85.7)	9 (42.9)	17 (84.5)	5 (26.3)
	C	4 (66.7)	1 (16.7)	12 (100.0) *	2 (16.7)
SDH:	W	2 (9.5)	9 (42.9)	1 (5.3)	10 (52.6)
	C	2 (33.3)	5 (83.3)	0 (0.0) *	6 (50.0)

W: summer measurement; C: winter measurement; CO: urine color; DH: dehydrated; H: hydrated; SDH: seriously dehydrated; USG: urine specific gravity. Variables are expressed as n (%). * *p* < 0.05 hydration status vs. team.

## Data Availability

The data presented in this study are available in the tables in this article. The data presented in this study are available on request from the corresponding author.

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
