# Peer review of "Analysis of Fluid Balance and Urine Values in Elite Soccer Players: Impact of Different Environments, Playing Positions, Sexes, and Competitive Levels"

_nutrients, 2024, doi:10.3390/nu16060903_

Round 1

Reviewer 1 Report

Comments and Suggestions for Authors

I have had the opportunity to review the paper entitled “Analysis of Fluid Balance and Urine in Elite Soccer Players: Impact of Different Environments, Playing position, Sex and Competitive Levels”. In this observational study, the authors assessed hydration status in elite soccer players. Body composition assessment is an important challenge nowadays and although the topic is generally worthy of consideration, it is my opinion several revisions are required. Detailed comments are listed below.

Introduction

-          Several key studies from the past 10 years regarding fluid assessment and hydration definition are missing. I would encourage to authors to read more around the subject as these aspects need to be considered in the introduction to form a basis for this paper. Also, the rationale for the conduct of this study is not very clear. I therefore suggest revising the introduction to make clear why your study is of importance for the field.

Methods

-          Please provide results for a power analysis that indicate your sample size was appropriate.

-          Stature data should be reported along with the body mass data.

-          Report if ISAK procedures were conducted by an accredited anthropometrist. Also, consider referring to a recent MDPI publication (Petri et al. 2024. ISAK-Based Anthropometric Standards for Elite Male and Female Soccer Players. https://doi.org/10.3390/ sports12030069). It is very important to highlight the importance if considering raw anthropometric data for comparison purposes as well as the importance to follow standardized procedures.

Results

-          The manuscript largely lacks illustrative representation by summarizing data in tables without showing the original data. This is a poor representation of scientific data. For instance, Figures 1 and 2 should show the individual data. Also, quality of the figure need to be improved.

Discussion

-          The discussion section is very descriptive and offers limited comparisons to previous research. Comparing anthropometric data with other soccer-related publications could represent a first tentative to address this gap. Furthermore, it seems like not properly hydration strategies among soccer players represent the main finding of this study. Similarly, how do practitioner benefit from that? What about the use of BIA for assessing hydration status in soccer? In the current form the rationale for the study is not clear, the new value is unclear, and I have difficulties finding specific take home messages for practitioners.

Other comments:

-          The text and tables are full of acronyms. Reading the manuscript becomes very difficult for a non-expert reader in the topic

Author Response

Introduction

-          Several key studies from the past 10 years regarding fluid assessment and hydration definition are missing. I would encourage to authors to read more around the subject as these aspects need to be considered in the introduction to form a basis for this paper. Also, the rationale for the conduct of this study is not very clear. I therefore suggest revising the introduction to make clear why your study is of importance for the field.

Response of the authors: Thank you for this comment. The introduction has been updated to include information justifying the study's objective, defining euhydration, and specifying factors that can affect dehydration.   However, we believe that the references used in the introduction are sufficiently important. These references include recent articles on soccer such as 10.1136/bjsports-2019-101961, 10.1519/SSC.0000000000000533, and 10.3390/nu14153188, as well as specific articles related to hydration, such as 10.3390/nu11071550 or 10.3305/nh.2014.29.1.6775, and the American College of Sports Medicine Position Stands. Our goal is to optimize the introduction's readability by keeping it short. However, if the reviewer considers that we need to introduce a study that we have not mentioned, we are open to modify it.

Methods

-          Please provide results for a power analysis that indicate your sample size was appropriate.

Response of the authors: Following the reviewer's suggestions, the information regarding the sample size has been provided.

-          Stature data should be reported along with the body mass data.

Response of the authors: We have added the height data of the players.

-          Report if ISAK procedures were conducted by an accredited anthropometrist. Also, consider referring to a recent MDPI publication (Petri et al. 2024. ISAK-Based Anthropometric Standards for Elite Male and Female Soccer Players. https://doi.org/10.3390/ sports12030069). It is very important to highlight the importance if considering raw anthropometric data for comparison purposes as well as the importance to follow standardized procedures.

Response of the authors: Thank you for this recommendation. The information provided is justified based on the measurements taken and the system used for measurement (ISAK).

Results

-          The manuscript largely lacks illustrative representation by summarizing data in tables without showing the original data. This is a poor representation of scientific data. For instance, Figures 1 and 2 should show the individual data. Also, quality of the figure need to be improved.

Response of the authors: Thank you for this recommendation. However, we respectfully disagree, as we believe that individual data does not provide essential information. Furthermore, the inclusion of such data may make it difficult for the reader to understand the figures. The quality of the figures has been improved (The high-resolution figures have been uploaded separately from the article to ensure that the publisher can assemble them without losing quality).

Discussion

-          The discussion section is very descriptive and offers limited comparisons to previous research. Comparing anthropometric data with other soccer-related publications could represent a first tentative to address this gap. Furthermore, it seems like not properly hydration strategies among soccer players represent the main finding of this study. Similarly, how do practitioner benefit from that? What about the use of BIA for assessing hydration status in soccer? In the current form the rationale for the study is not clear, the new value is unclear, and I have difficulties finding specific take home messages for practitioners.

Response of the authors: We have added point 4.4 practical applications to give more practicality to our study. In addition, we have added information regarding the potential use of BIA for hydration assessment, as well as its disadvantages. In relation to the comparison of anthropometric data, we acknowledge the significance and relevance of such data. However, we have noted that several published studies that assessed hydration did not provide any anthropometric data beyond weight and height. Therefore, we have excluded any further investigation into fat mass and fat-free mass. Furthermore, the discussion is hindered by the use of mixed measurement methods (anthropometry and bioimpedance) and different anthropometric equations, which make it difficult to compare results. Our research group recommends against combining different measurement methods or formulas due to significant variations in values, as demonstrated in our previously published articles (10.3390/nu15051160, 10.3390/app13084782 and 10.3390/app132011441).

Other comments:

-          The text and tables are full of acronyms. Reading the manuscript becomes very difficult for a non-expert reader in the topic

Response of the authors: We acknowledge the feedback on improving the text. However, we understand that the use of abbreviations, particularly n-RM (Non-repeated measurements) and RM (Repeated measurements), may simplify the article and enhance readability. We recognize that this decision may affect the text, but we deemed it necessary to conduct statistical analysis using both repeated and non-repeated measurements to minimize inter-individual error and increase precision. Additionally, we do not believe that the use of six abbreviations will have a negative impact on the article's progress.

Reviewer 2 Report

Comments and Suggestions for Authors

This study is expected to provide appropriate information to maintain appropriate conditioning of soccer players by analyzing changes in fluid balance according to environmental conditions and training content of soccer players. However, correction and supplementation of the following matters are required.

1. In order to analyze the effects of gender, environmental conditions, training intensity, etc., I believe that the reliability of the results is reduced due to the discrepancy in the number of each target type. Appropriate defense or discussion on this matter is required.

2. The resolution of Figures 1 and 2 is believed to be significantly low. A clearer presentation is required, and in the case of Figure 2, the standard deviation is also requested.

3. At the end of the discussion, please add a comment on how the results of this study can be used in the training field.

Author Response

  1. In order to analyze the effects of gender, environmental conditions, training intensity, etc., I believe that the reliability of the results is reduced due to the discrepancy in the number of each target type. Appropriate defense or discussion on this matter is required.

Response of the authors: Following the reviewer's suggestions, the information regarding the sample size has been provided. As stated in the limitations, the sample size was small. However, based on the sample size highlighted in section 2.1 study design, the required sample size was 62 soccer players. Initially, all soccer players were included in the study, but due to reasons beyond the research group's control (such as injured players, those who withdrew from training due to physical discomfort, or menstruation in women), the number of soccer players was reduced. Finally, a strong point of the sample is that all participants come from the same soccer club, train on the same fields, experience similar weather conditions, follow the same work methodology as the technical and medical staff of the club, and have the same level of intensity (MD-3) during competitions.

  1. The resolution of Figures 1 and 2 is believed to be significantly low. A clearer presentation is required, and in the case of Figure 2, the standard deviation is also requested.

Response of the authors: Thank you very much for your kind comment, but we do not understand what the reviewer mean. Figure 2 reflects the percentages of subjects who had an adequate hydration status, a dehydration state, or a severe dehydration state. The standard deviation of a percentage cannot be calculated. The quality of the figures has been improved (The high-resolution figures have been uploaded separately from the article to ensure that the publisher can assemble them without losing quality).

  1. At the end of the discussion, please add a comment on how the results of this study can be used in the training field.

Response of the authors: We have added point 4.4 practical applications to give more practicality to our study.
